# How Gradient Descent Separates Data with Neural Collapse: A Layer-Peeled Perspective

## Abstract

In this paper, we study the inductive bias of the neural features and parameters from neural networks with cross-entropy loss. We study a surrogate model named unconstrained layer peeled model (ULPM), which helps us to illustrate that the features and classifiers in the last layer of the neural network will converge to a certain neural collapse structure [28], where the cross-example within-class variability of the last-layer features collapse to zero and the class-means converge to a Simplex Equiangular Tight Frame (ETF). We illustrate that the ULPM with cross-entropy loss enjoys a benign global landscape on this model where all the critical points are strict saddle points except the only global minimizers which exhibit neural collapse phenomenon. Empirically we show that our results also hold during the training of neural networks in real world tasks when explicit regularization or weight decay is not included.

## 1 Introduction

Deep learning has achieved state-of-the-art performances in various applications [20], from computer vision [16], to natural language processing[6] and even scientific discovery [23, 41]. Despite the empirical successes achieved, how gradient descent or its variants leads deep neural networks to be biased towards solutions with good generalization performance on the test set is still a major open question. To develop a theoretical foundation for deep learning, many works have studied the implicit bias of gradient descent in different settings [21, 1, 37, 33, 25, 3].

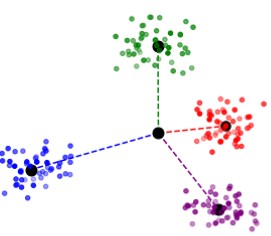

Figure 1: Illustration of Neural Collapse [28].

It is well-acknowledged that well-trained end-to-end deep architectures have the ability to effectively extract features relevant to the given label. Although theoretical analysis of deep learning has several achievements in recent years [2, 13], most of the works that aim to analyze properties of the final output function fail to understand the feature learned. Recently in [28], authors observe that the within-class cross-sample features will collapse to the mean and the mean will converge to an Equiangular Tight Frame (ETF) during the terminal phase of training, *i.e.* after achieving zero training error and interpolating the in-sample training data. Such phenomenon, namely Neural Collapse (NC) [28], provides a clear view of how the last layer features in the neural network involve after interpolation and enables us to understand the benefit of training after achieving zero training error to achieve better properties in generalization and robustness. To theoretically analyze the neuron

collapse phenomenon, [9, 24, 39] propose the Layer-Peeled Model (LPM) as a simplification for neural networks, where the last-layer features are modeled as free optimization variables. In particular, in a $K$-class classification problem using a neural network with $d$ neurons in the last hidden layer, a corresponding class of LPMs can be defined through the form

$$\min_{\boldsymbol{W}, \boldsymbol{H}} \frac{1}{N} \sum_{i=1}^{N} \mathcal{L} \left(\boldsymbol{W} \boldsymbol{h}_i, \boldsymbol{y}_i\right)$$

$$\text{s.t. } \frac{1}{2} \|\boldsymbol{W}\|_F^2 \leq C_1, \frac{1}{2} \|\boldsymbol{H}\|_F^2 \leq C_2 \tag{1}$$

for some positive constant $C_1, C_2$. Here $\boldsymbol{W} = [\boldsymbol{w}_1, \boldsymbol{w}_2, \cdots, \boldsymbol{w}_K]^\top \in \mathbb{R}^{K \times d}$ is the weight of the final linear classifier, $\boldsymbol{H} = [\boldsymbol{h}_1, \boldsymbol{h}_2, \cdots, \boldsymbol{h}_N] \in \mathbb{R}^{d \times N}$ is the feature of the last layer and $y_i$ is the corresponding label. The intuition behind LPM is that the modern deep networks are often highly over-parameterized, with the capacity to learn any representations of the input data. It has been shown that equiangular tight frame (ETF), *i.e.* feature with neural collapse, is the only global optimum of the LPM objective (1) [9, 24, 39]. However, even for this simplified model, the non-convexity nature of it makes the analysis highly non-trivial. In this paper we aim to understand *how gradient descent separates data with neural collapse*. To do this, we build a connection between the neural collapse with the recently proposed normalized margin [25, 38]. In [25], the authors shows that, using gradient descent, the direction of the weight converges to the direction that maximizes the $\ell_2$-margin of the data while the norm of the weight diverges to $+\infty$ in homogeneous neural networks. Based on these results, we introduce neural collapse margin and use it provide a convergence result to the first order stationary point of the minimum-norm separation problem. Furthermore, we illustrate that the cross-entropy loss enjoys a benign global landscape where all the critical points are strict saddles in the tangent space except the only global minimizers which exhibit neural collapse phenomenon. The analysis provides insights on how gradient descent separates data during the training of neural networks with neural collapse and the benefit of training after interpolation on generalization and robustness. We verify our insights via empirical experiments.

| Reference | Contribution | Feature Norm Constraint | Feature Norm Regularization | Loss Function |
|---|---|---|---|---|
| [28] | Empirical Results | ✗ | ✗ | Cross-Entropy Loss |
| [9] | Global Optimum | ✔ | ✗ | Cross-Entropy Loss |
| [39] | Global Optimum | ✔ | ✗ | Cross-Entropy Loss |
| [24] | Global Optimum | ✔ | ✗ | Cross-Entropy Loss |
| [26] | Training Dynamics | ✗ | ✗ | $\ell_2$ Loss |
| [43] | Landscape Analysis | ✗ | ✔ | Cross-Entropy Loss |
| This paper | Training Dynamics+ Landscape Analysis | ✗ | ✗ | Cross-Entropy Loss |

Table 1: Comparison of Recent Analysis for Neural Collapse. We provide strongest theoretical results with minimum modification on the training objective function.

Besides, [26] and a concurrent paper [43] also provide landscape and optimization analysis to study neural collapse phenomenon, we summarize the connection and difference with our paper in Table 1. Our result doesn't introduce any extra feature norm constraint or feature norm regularization, which are not commonly used in the realistic deep learning. We put the detailed discussion in Section 5.2.

## 1.1 Contribution

We summarize our contribution as follows.

- We build a relationship between the max-margin analysis [33, 27, 25] with the neural collapse and provide the inductive bias analysis to the feature rather than the output function.

- Previous works only prove that Gradient Descent on homogeneous neural networks will converge to the KKT point of the corresponding minimum-norm separation problem. However, the minimum-norm separation problem is still a highly non-convex problem. In this paper, we prove that the ULPM cases enjoys a benign landscape and characterize the neural collapse property of the global minimizer.

- We show that although the gradient descent on cross entropy loss will push the parameters to infinity, the landscape in the tangent space has no spurious minimum thus many optimization algorithms will converge only along the neural collapse directions .

## 1.2 Related Work

**Inductive Bias of Gradient Descent:** To understand how gradient or its variants descent helps deep learning to find solutions with good generalization performance on the test set. A recent line of research have studied the implicit bias of gradient descent in different settings. As example, gradient descent is biased towards model have smaller weight [21, 1, 37] and will converge to large margin solution [33, 27, 25, 7, 14] while using logistic loss. For linear networks, [3, 31, 12] have shown that gradient descent will find out a low rank approximation.

**Loss Landscape Analysis:** Although the practical optimization problems encountered in machine learning are often nonconvex, recent works have shown that critical points other than the good ones always lies in the balanced superpositions of symmetric copies of the ground truth according to the hidden symmetries in the objective function [34, 42] which leads to a benign global landscape. In particular, these landscapes do not exhibit spurious local minimizers or flat saddles and can be optimized easily via gradient based methods [10]. The examples including phase retrieval [36], low-rank matrix recovery [11, 10], dictionary learning [35, 30, 19], blind deconvolution [18].

# 2 Preliminaries and Problem Setup

## 2.1 Preliminaries

We consider a dataset with $K$ classes: $\bigcup_{k=1}^{K} \{\boldsymbol{x}_{k,i}\}_{i=1}^{n_k}$. For simplicity, we assume the dataset is balanced, *i.e.* $n_1 = \cdots = n_K = n$. A standard fully connected neural network can be represented as:

$$f(\boldsymbol{x}; \boldsymbol{W}_{full}) = \boldsymbol{b}_L + \boldsymbol{W}_L \sigma(\boldsymbol{b}_{L-1} + \boldsymbol{W}_{L-1} \sigma(\cdots \sigma(\boldsymbol{b}_1 + \boldsymbol{W}_1 \boldsymbol{x}))). \tag{2}$$

Here $\boldsymbol{W}_{full} = (\boldsymbol{W}_1, \boldsymbol{W}_2, \cdots, \boldsymbol{W}_L)$ denote the weight matrices in each layer and $(\boldsymbol{b}_1, \boldsymbol{b}_2, \cdots, \boldsymbol{b}_L)$ are the bias terms, $\sigma(\cdot)$ stands for the nonlinear activation function, for example, ReLU or sigmoid. Let $\boldsymbol{h}_{k,i} = \sigma(\boldsymbol{b}_{L-1} + \boldsymbol{W}_{L-1} \sigma(\cdots \sigma(\boldsymbol{b}_1 + \boldsymbol{W}_1 \boldsymbol{x}_{k,i}))) \in \mathbb{R}^d$ denote the last layer feature for data $\boldsymbol{x}_{k,i}$ and $\bar{\boldsymbol{h}}_k = \frac{1}{n} \sum_{i=1}^{n} \boldsymbol{h}_{k,i}$ the feature mean within in the k-th class. Without loss of generality, we can absorb the bias term into the weight matrix by adding a scalar into each feature vectors, so we will ignore the bias term in the following analysis. Let $\boldsymbol{W} \in \mathbb{R}^{K \times d} = \boldsymbol{W}_L = [\boldsymbol{w}_1, \boldsymbol{w}_2, \cdots, \boldsymbol{w}_K]^\top$ be the weight of the final linear classifier. Neural collapse is the phenomenon that the final layer feature will convergence to a simplex equiangular tight frame (ETF):

**Definition 2.1.** A symmetric matrix $\boldsymbol{M} \in \mathbb{R}^{K \times K}$ is said to be simplex equiangular tight frame (ETF) if

$$\boldsymbol{M} = \sqrt{\frac{K}{K-1}} \boldsymbol{Q}(\boldsymbol{I}_K - \frac{1}{K} \boldsymbol{1}_K \boldsymbol{1}_K^\top). \tag{3}$$

Where $\boldsymbol{Q} \in \mathbb{R}^{K \times K}$ is an orthogonal matrix.

The four criteria of neural collapse can be formulated precisely as

- **(NC1) Variability collapse:** As training progresses, the within-class variation of the activation becomes negligible as these activation collapse to their class-means $\bar{\boldsymbol{h}}_k = \frac{1}{n} \sum_{i=1}^{n} \boldsymbol{h}_{k,i}$.

$$||\boldsymbol{h}_{k,i} - \bar{\boldsymbol{h}}_k|| = 0, \quad \forall 1 \leq k \leq K$$

- **(NC2) Convergence to Simplex ETF:** The vectors of the class-means (after centering by their global-mean converge to having equal length, forming equal-sized angles between any given pair, and being the maximally pairwise-distanced configuration constrained to the previous two properties.

$$cos(\bar{\boldsymbol{h}}_k, \bar{\boldsymbol{h}}_j) = -\frac{1}{K-1}, \quad ||\bar{\boldsymbol{h}}_k|| = ||\bar{\boldsymbol{h}}_j||, \quad \forall k \neq j$$

- **(NC3) Convergence to self-duality:** The linear classifiers and class-means will converge to each other, up to rescaling.

$$\exists C \text{ s.t. } \boldsymbol{w}_k = C\bar{\boldsymbol{h}}_k, \quad \forall 1 \leq k \leq K$$

- **(NC4) Simplification to Nearest Class-Center** For a given deepnet activation $\boldsymbol{h} = \sigma\left(\boldsymbol{b}_{L-1} + \boldsymbol{W}_{L-1}\sigma\left(\cdots\sigma\left(\boldsymbol{b}_1 + \boldsymbol{W}_1\boldsymbol{x}\right)\right)\right) \in \mathbb{R}^d$, the network classifier converges to choose whichever class has the nearest train class-mean

$$\arg\min_k \langle\boldsymbol{w}_k, \boldsymbol{h}\rangle \to \arg\min_k \left\|\boldsymbol{h} - \bar{\boldsymbol{h}}_k\right\|,$$

In this paper, we say a point $\boldsymbol{W} \in \mathbb{R}^{K \times d}, \boldsymbol{H} \in \mathbb{R}^{d \times nK}$ satisfies neural collapse conditions or is neural collapse solution if these four criteria are all satisfied for $(\boldsymbol{W}, \boldsymbol{H})$.

## 2.2  Problem Setup

In this paper, we mainly focus on the neural collapse phenomenon, which is only related to the classifiers and features in the last layer. Since general analysis on the highly non-smooth and non-convex neural network is difficult, here we peel down the last layer of neural network and propose the following **Unconstrained Layer-Peeled Model (ULPM)** as a simplification to capture the main characteristic related to neural collapse during the training dynamics. Similar simplification is common used in previous theoretical works [24, 9, 39, 43], but ours don't have any constraint or regularization on features and stands closer to realistic neural network models. We need to mention that although [26] also study the unconstrained model, their analysis is highly dependent on the $\ell_2$ loss function which is rarely used in classification task while ours can address the most popular cross entropy loss.

Let $\boldsymbol{W} = [\boldsymbol{w}_1, \boldsymbol{w}_2, \cdots, \boldsymbol{w}_K]^\top \in \mathbb{R}^{K \times d}$ and $\boldsymbol{H} = [\boldsymbol{h}_{1,1}, \cdots, \boldsymbol{h}_{1,N}, \boldsymbol{h}_{2,1}, \cdots, \boldsymbol{h}_{K,N}] \in \mathbb{R}^{d \times KN}$ be the matrices of classifiers and features in the last layer, where $K$ is the number of classes and $N$ is the number of data points in each classes. The Unconstrained Layer-Peeled Model is defined as following:

$$\min_{\boldsymbol{W}, \boldsymbol{H}} \ \mathcal{L}(\boldsymbol{W}, \boldsymbol{H}) = -\sum_{k=1}^{K}\sum_{i=1}^{n} \log\left(\frac{\exp(\boldsymbol{w}_k^\top \boldsymbol{h}_{k,i})}{\sum_{j=1}^{K}\exp(\boldsymbol{w}_j^\top \boldsymbol{h}_{k,i})}\right) \tag{4}$$

Here we do not have any constrain or regularization on features, which corresponds to the absence of weight decay in deep learning training. The objective function (4) is generally non-convex on $(\boldsymbol{W}, \boldsymbol{H})$ and we aim to study the landscape of the objective function (4). Furthermore, we consider the gradient flow of the the objective function

$$\frac{d\boldsymbol{W}(t)}{dt} = \frac{\partial\mathcal{L}(\boldsymbol{W}(t), \boldsymbol{H}(t))}{\partial\boldsymbol{W}}, \frac{d\boldsymbol{H}}{dt} = \frac{\partial\mathcal{L}(\boldsymbol{W}(t), \boldsymbol{H}(t))}{\partial\boldsymbol{H}}.$$

We also trace the the dynamic of the loss function $\mathcal{L}(t) := \mathcal{L}(\boldsymbol{W}(t), \boldsymbol{H}(t))$ and study the convergence of $(\boldsymbol{W}(t), \boldsymbol{H}(t))$.

**Notations.** We denote $||\cdot||_F$ the Frobenius norm, $\|\cdot\|_2$ the matrix spectral norm, $\|\cdot\|_*$ the nuclear norm, $\|\cdot\|$ the vector $l_2$ norm and $tr(\cdot)$ the trace of matrices. We use $[K] := \{1, 2, \cdots, K\}$ to denote the set of indices up to $K$.

## 3  Main Results

In this section, we present our main results about the training dynamics and landscape analysis about (4). We organize the section as follows: First in Section 3.1.1, we show the relationship between margin and neural collapse in our surrogate model. Inspired by this relationship, we propose a minimum-norm separation problem (5) and show the connection between the convergence direction of gradient flow and the KKT point of (5). In addition, we explicitly solve the global optimum of (5) and show it must satisfy neural collapse conditions. However, due to the non-convexity, we find an Example 3.1 in Section 3.2 which shows that there exist some bad KKT points such that simple gradient flow will get stuck in them and not converge to neural collapse solution which is proved to be optimal in Theorem 3.3. Then we present our second–order analysis result in Theorem 3.4 to show that those bad points will exhibit decreasing directions in the tangent space thus if we add some noise in the training algorithm (e.g. use stochastic gradient descent), our algorithm can escape from those directions and can only converge to the neural collapse solutions.

## 3.1 Convergence To The First–Order Stationary Point

### 3.1.1 Neural Collapse Margin

Before we state our convergence result, let's first discuss the relationship between margin and neural collapse. By building the relationship between them we can have a better intuition about why gradient flow can converge to neural collapse solution since the convergence to max-margin solutions has been studied in many literature [21, 25, 1, 37]. Recall the margin of a single data point $x_{k,i}$ and associated feature $h_{k,i}$ as $q_{k,i}(W, H) := w_k^\top h_{k,i} - \max_{j \neq k} w_j^\top h_{k,i}$. [5, 4]. To bridge the margin theory with neural collapse phenomenon, we define the following neural collapse margin:

**Definition 3.1.** We define the the **Neural Collapse Margin** for the entire dataset as $q_{\min}(W, H) = \min_{k \in [1,K], i \in [1,n]} q_{k,i}(W, H)$.

The following lemma shows that the neural collapse margin is an indicator of the neural collapse phenomenon in the sense that collapsed margin minimize the neural collapse margin. Thus we can trace the neural collapse margin to study the convergence to the neural collapse solution.

**Lemma 3.1** (Neural Collapse Margin as an Indicator of Neural Collapse). *The neural collapse margin always smaller than*

$$q_{\min}(W, H) \leq \frac{\|W\|_F^2 + \|H\|_F^2}{2(K-1)\sqrt{n}}$$

*and $(W, H)$ must satisfies the neural collapse conditions when the inequality above is reduced to an equality.*

### 3.1.2 Convergence Results

Now we present our result about the convergence of gradient flow on the ULPM (4). Following [25], we link gradient flow on cross-entropy loss with a minimum-norm separation problem.

**Theorem 3.1.** *For problem (4), let $(W(t), H(t))$ be the path of gradient flow at time t, if there exist a time $t_0$ such that $\mathcal{L}_{CE}(W(t_0), H(t_0)) < \log 2$, then any limit point of $\{(\hat{H}(t), \hat{W}(t)) := (\frac{H(t)}{\sqrt{\|W(t)\|_2^2 + \|H(t)\|_2^2}}, \frac{W(t)}{\sqrt{\|W(t)\|_2^2 + \|H(t)\|_2^2}})\}$ is along the direction of an Karush-Kuhn-Tucker (KKT) point of the following minimum-norm separation problem:*

$$\min_{W,H} \frac{1}{2}\|W\|_F^2 + \frac{1}{2}\|H\|_F^2 \tag{5}$$
$$s.t. \quad \forall k \neq j \in [K], i \in [n], \quad w_k^\top h_{k,i} - w_j^\top h_{k,i} \geq 1.$$

*Remark* 3.1. Indeed, the problem (5) can be reorganized to maximize neural collapse margin such that the norm is constrained to be lower than a certain value. The proof is as follows, for all feasible solutions $(W, H)$, we can find that $\forall \alpha \geq q_{min}(W, H)^{-1/2}, \alpha(W, H)$ are still feasible thus the minimum objective value is $\frac{\frac{1}{2}\|W\|_F^2 + \frac{1}{2}\|H\|_F^2}{q_{min}(W,H)^{1/2}}$ along the direction of $(W, H)$. Then take minimum among all the directions we can find the minimum is attained if and only if $(W, H)$ attains the maximum neural collapse margin on the sphere $\{(W, H) : \|W\|_F^2 + \|H\|_F^2 \leq C\}$

The Theorem 3.1 indicates that the convergent direction of gradient flow is restricted to those max-margin directions, which usually enjoy some good properties on robustness or generalization performance. Generally speaking, the KKT conditions are not sufficient to obtain global optimality since the minimum-norm separation problem (5) is non-convex. Moreover, in some certain occasions, KKT conditions may be even not necessary for global optimum. However, we can have a precise characterization about the optimum from another perspective, the following result shows that the global optimum of this problem satisfies neural collapse conditions.

**Theorem 3.2.** *Every global optimum of the minimum-norm separation problem (5) is also a KKT point and it satisfies the neural collapse conditions.*

To illustrate how does (5) related to (4) and gain insight about Theorem 3.1, we provided the following lemmas to show that when t is sufficient large, the $(W(t), H(t))$ is an $(\epsilon, \delta)$ approximate KKT point after appropriate scaling, where the $(\epsilon, \delta)$ converges to zero when $t \to \infty$. Then as shown in [8] we know that the limit of these $(\epsilon, \delta)$ approximate KKT point is exact KKT point. Detailed definition of KKT points and approximate KKT points can be found in appendix.

**Lemma 3.2.** *If there exist a time $t_0$ such that $\mathcal{L}(\boldsymbol{W}(t_0), \boldsymbol{H}(t_0)) < \log 2$, then for any $t > t_0$ $(\tilde{\boldsymbol{W}}(t), \tilde{\boldsymbol{H}}(t)) := (\boldsymbol{W}(t), \boldsymbol{H}(t))/q_{\min}(\boldsymbol{W}(t), \boldsymbol{H}(t))^{1/2}$ is a $(\epsilon, \delta)$ - approximate KKT point of the following minimum-norm separation problem. More precisely, we have*

$$\epsilon = \sqrt{\frac{2(1 - \beta(t))}{C}}, \delta = \frac{K}{2Cq_{min}(t)}$$

*where:*

$$\beta = \frac{tr(\boldsymbol{W}^\top \nabla_{\boldsymbol{W}} \mathcal{L}(\boldsymbol{W}, \boldsymbol{H})) + tr(\boldsymbol{H}^\top \nabla_{\boldsymbol{H}} \mathcal{L}(\boldsymbol{W}, \boldsymbol{H}))}{\sqrt{||\boldsymbol{W}||_F^2|| + ||\boldsymbol{H}||_F^2} \sqrt{||\nabla_{\boldsymbol{W}} \mathcal{L}(\boldsymbol{W}, \boldsymbol{H})||_F^2|| + ||\nabla_{\boldsymbol{H}} \mathcal{L}(\boldsymbol{W}, \boldsymbol{H})||_F^2}}$$

*is the angle between $(\boldsymbol{W}, \boldsymbol{H})$ and its corresponding gradient and $C$ is a positive constant.*

**Lemma 3.3.** *If there exist a time $t_0$ such that $\mathcal{L}_{CE}(\boldsymbol{W}(t_0), \boldsymbol{H}(t_0)) < \log 2$, then we have:*

$$\beta(t) \to 1, \quad q_{min}(t) \to \infty \text{ as } t \to \infty \tag{6}$$

*which implies that $\epsilon \to 0$ and $\delta \to 0$ when time $t$ goes to infinity.*

### 3.2 Second–Order Landscape Analysis

Due to the non-convex nature of the objective (4), we can't achieve such global solution efficiently. The global optimality condition shown in Theorem 3.2 still can't guarantee convergence to neural collapse. In this section, we aim to show that this non-convex optimization problem is actually not scary.

Different from previous landscape analysis of non-convex problem, where people aim to show that the objective has a negative directional curvature around any stationary point [34, 42], once features can be perfectly separated, the ULPM objective (4) will always decrease along the direction of the current point and the optimum is attained only in infinity. Although growing along all of those perfectly separation directions can let the loss function decreasing to 0, the speed of decreasing are quite different and there exists an optimal direction with fastest decreasing speed. However, simple first–order analysis may fail to interpret how does gradient flow move among these directions and we need second–order analysis to help us fully characterize the realistic training dynamics. Here is an example illustrating our motivation.

**Example 3.1** (A Motivating Example). Consider the case when $K = 4, n = 1$, let $(\boldsymbol{W}, \boldsymbol{H})$ be the following point:

$$\boldsymbol{W} = \boldsymbol{H} = C \begin{bmatrix} 1 & -1 & 0 & 0 \\ -1 & 1 & 0 & 0 \\ 0 & 0 & 1 & -1 \\ 0 & 0 & -1 & 1 \end{bmatrix} \tag{7}$$

One can easily verify that this $(\boldsymbol{W}, \boldsymbol{H})$ enables our model to classify all of the features perfectly. Further more, we can show it is along the direction of a KKT point of the minimum-norm separation problem (5) by construct the Lagrangian multiplier $\Lambda = (\lambda_{ij})_{i,j=1}^K$ as following:

$$\Lambda = \begin{bmatrix} 0 & 0 & \frac{1}{2} & \frac{1}{2} \\ 0 & 0 & \frac{1}{2} & \frac{1}{2} \\ \frac{1}{2} & \frac{1}{2} & 0 & 0 \\ \frac{1}{2} & \frac{1}{2} & 0 & 0 \end{bmatrix} \tag{8}$$

And the gradient of $(\boldsymbol{W}, \boldsymbol{H})$ is

$$\nabla_{\boldsymbol{W}} \mathcal{L}(\boldsymbol{W}, \boldsymbol{H}) = \nabla_{\boldsymbol{H}} \mathcal{L}(\boldsymbol{W}, \boldsymbol{H}) = -C \frac{2 + 2e^{-2C^2}}{2 + 2e^{-2C^2} + 2e^{2C^2}} \begin{bmatrix} 1 & -1 & 0 & 0 \\ -1 & 1 & 0 & 0 \\ 0 & 0 & 1 & -1 \\ 0 & 0 & -1 & 1 \end{bmatrix} \tag{9}$$

We can find that the directions of gradient and the parameter align with each other (*i.e.* $\boldsymbol{W} // \nabla_{\boldsymbol{W}} \mathcal{L}(\boldsymbol{W}, \boldsymbol{H}), \boldsymbol{H} // \nabla_{\boldsymbol{H}} \mathcal{L}(\boldsymbol{W}, \boldsymbol{H})$), which implies simple gradient descent get stuck in this direction and only grow the parameter norm. However, if we construct:

$$\boldsymbol{W}' = \boldsymbol{H}' = C \begin{bmatrix} 1 & \alpha & \beta & \beta \\ \alpha & 1 & \beta & \beta \\ \beta & \beta & 1 & \alpha \\ \beta & \beta & \alpha & 1 \end{bmatrix}, \quad \alpha^2 + 2\beta^2 = 1, \alpha < 0, \beta < 0 \tag{10}$$

Then $\forall \epsilon > 0$, we can choose appropriate $\alpha, \beta$ such that (see detailed computation in Appendix):

$$||\boldsymbol{W}'||_F^2 = ||\boldsymbol{W}||_F^2, ||\boldsymbol{H}'||_F^2 = ||\boldsymbol{H}||_F^2,$$
$$||\boldsymbol{W}' - \boldsymbol{W}||_F^2 + ||\boldsymbol{H}' - \boldsymbol{H}||_F^2 < \epsilon, \mathcal{L}(\boldsymbol{W}', \boldsymbol{H}') \leq \mathcal{L}(\boldsymbol{W}, \boldsymbol{H}) \quad (11)$$

The results in (11) indicate that $(\boldsymbol{W}', \boldsymbol{H}')$ is a saddle point on the sphere and there exists many better direction $(\boldsymbol{W}', \boldsymbol{H}')$ staying very close to the original direction $(\boldsymbol{W}, \boldsymbol{H})$. Although simple gradient descent will always move along the original direction, once we add some noise in the training (e.g. stochastic gradient descent), the optimization algorithm can find this better direction and escape the original bad direction.

In Example 3.1, we show that there does exist some suboptimal KKT point of the minimum-norm separation problem (5), but there also exist some better points close to it thus stochastic gradient method can easily escape form them. In the following theorem, we will show that the best directions are neural collapse solutions in the sense that the loss function is lowest among all the growing directions.

**Theorem 3.3.** *The optimal value of loss function (4) on a sphere is attained* (i.e. $\mathcal{L}(\boldsymbol{W}, \boldsymbol{H}) \leq \mathcal{L}(\boldsymbol{W}', \boldsymbol{H}'), \forall ||\boldsymbol{W}'||_F^2 + ||\boldsymbol{H}'||_F^2 = ||\boldsymbol{W}||_F^2 + ||\boldsymbol{H}||_F^2$) *if only if the $(\boldsymbol{W}, \boldsymbol{H})$ satisfies neural collapse conditions and $||\boldsymbol{W}||_F = ||\boldsymbol{H}||_F$.*

*Remark* 3.2. Note that the second conditions is necessary since neural collapse conditions don't specify the norm ratio of $\boldsymbol{W}$ and $\boldsymbol{H}$. That is, if $(\boldsymbol{W}, \boldsymbol{H})$ satisfies neural collapse conditions, $(\alpha \boldsymbol{W}, \beta \boldsymbol{H}), \forall \alpha, \beta \in \mathbb{R}$ will also satisfies them but only some certain $\alpha, \beta$ are optimal.

Now we turns to those points that don't satisfy neural collapse conditions. To formalize our discussion in the motivating Example 3.1, we first introduce the tangent space:

**Definition 3.2** (tangent space)**.** The tangent space of $(\boldsymbol{W}, \boldsymbol{H})$ is defined to be a set of directions that are orthogonal to $(\boldsymbol{W}, \boldsymbol{H})$ :

$$\mathcal{T}(\boldsymbol{W}, \boldsymbol{H}) = \{\Delta \boldsymbol{W} \in \mathbb{R}^{K \times d}, \Delta \boldsymbol{H} \in \mathbb{R}^{d \times nK}) : tr(\boldsymbol{W}^\top \Delta \boldsymbol{W}) + tr(\boldsymbol{H}^\top \Delta \boldsymbol{H}) = 0\} \quad (12)$$

Our next result justify our observation in the Example 3.1 that for every suboptimal points, there exist a direction in the tangent space such that move along this direction will leads to a lower objective value.

**Theorem 3.4.** *If $(\boldsymbol{W}, \boldsymbol{H})$ is not the optimal solutions in Theorem 3.3, then $\exists (\Delta \boldsymbol{W}, \Delta \boldsymbol{H}) \in \mathcal{T}(\boldsymbol{W}, \boldsymbol{H}), M > 0$ such that*

$$\forall 0 < \delta < M, \mathcal{L}(\boldsymbol{W} + \delta \Delta \boldsymbol{W}, \boldsymbol{H} + \delta \Delta \boldsymbol{H}) \leq \mathcal{L}(\boldsymbol{W}, \boldsymbol{H}) \quad (13)$$

*. Further more, it implies that $\forall \epsilon > 0, \exists (\boldsymbol{W}', \boldsymbol{H}')$ such that:*

$$||\boldsymbol{W}'||_F^2 + ||\boldsymbol{H}'||_F^2 = ||\boldsymbol{W}||_F^2 + ||\boldsymbol{H}||_F^2,$$
$$||\boldsymbol{W}' - \boldsymbol{W}||_F^2 + ||\boldsymbol{H}' - \boldsymbol{H}||_F^2 < \epsilon, \mathcal{L}(\boldsymbol{W}', \boldsymbol{H}') \leq \mathcal{L}(\boldsymbol{W}, \boldsymbol{H}) \quad (14)$$

*Remark* 3.3. The result in (13) give us a decreasing direction orthogonal to the direction of $(\boldsymbol{W}, \boldsymbol{H})$, as shown in Example 3.1, the gradient might be parallel to $(\boldsymbol{W}, \boldsymbol{H})$, the decreasing direction must be obtained by analyze the Hessian matrices and it further indicates that these points are exactly saddle points in the tangent space, a formal statement and definition can be found in appendix. For a large family of stochastic optimization algorithm , the projection of noise onto this decreasing direction is not zero with probability 1, so its those algorithms will escape the bad point and no longer move along this direction within a small number of iterations.

## 4   Empirical Results

**Gradient Descent on the ULPM Objective.**   We first conduct experiments on the ULPM objective (4) to support the results of convergence towards Neural Collapse in our theories. We set $N = 10$, $K = 5, d = 20$ and use gradient descent with learning rate 5 to run $10^5$ epochs. We characterize the dynamics of the training procedure in Figure 2, through four aspects: (1) variation of the centered class-mean features' norms (*i.e.*, $\text{Std}(||\bar{\boldsymbol{h}}_k - \bar{\boldsymbol{h}}||)/\text{Avg}(||\bar{\boldsymbol{h}}_k - \bar{\boldsymbol{h}}||)$) and the variation of the classifier's norms (*i.e.*, $\text{Std}(||\bar{\boldsymbol{w}}_k||)/\text{Avg}(||\bar{\boldsymbol{w}}_k||)$). (2) Within-class variation of last layer features (*i.e.*, $\text{Avg}(||\boldsymbol{h}_{k,i} - \boldsymbol{h}_k||)/\text{Avg}(||\boldsymbol{h}_{k,i} - \bar{\boldsymbol{h}}||)$). (3) The cosines between pairs of last layer features (*i.e.*,

Avg($|\cos(\bar{\boldsymbol{h}}_k, \bar{\boldsymbol{h}}_{k'}) + 1/(K-1)|$)) and that of the classifiers (*i.e.*, Avg($|\cos(\bar{\boldsymbol{w}}_k, \bar{\boldsymbol{w}}_{k'}) + 1/(K-1)|$)). (4) The distance between normalized centered classifier and normalized last layer feature (*i.e.*, Avg($|(\bar{\boldsymbol{h}}_k - \bar{\boldsymbol{h}})/\|\bar{\boldsymbol{h}}_k - \bar{\boldsymbol{h}}\| - \bar{\boldsymbol{w}}_k/\|\bar{\boldsymbol{w}}_k\||$)). Empirically we observe that logarithm of the two variations of norms (in the first aspect) decrease approximately at rate $O(1/(\log(t)))$, and the remaining quantities decrease approximately at rate $O(1/(\log(t)))$.

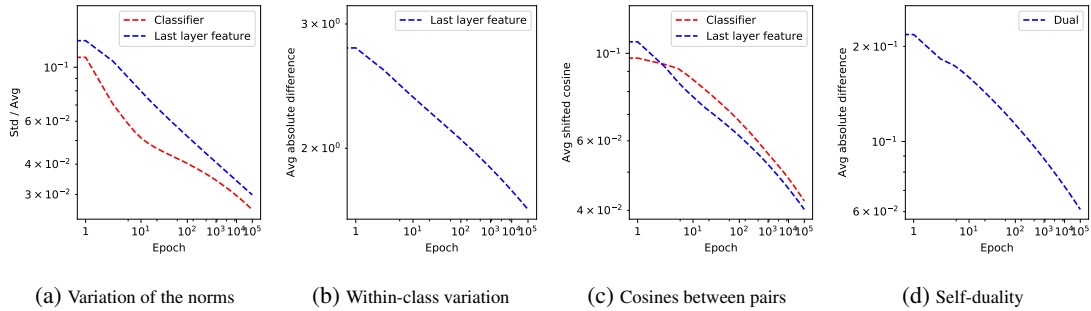

(a) Variation of the norms      (b) Within-class variation      (c) Cosines between pairs      (d) Self-duality

Figure 2: Training dynamics in ULPM. The $x$-axis in the figures are set to have $\log(\log(t))$ scales and the $y$-axis in the figures are set to have log scales. (a) The dynamics of the variation of the centered class-mean features' norms (shown in blue) and the variation of the classifier's norms (shown in red). We observe that the logarithm of both terms decrease at rate $O(1/(\log(t)))$. (b) The dynamics of the within-class variation of last layer features. Logarithm of the variation converge approximately at rate $O(1/\log(t))$. (c) The dynamics of the cosines between pairs of last layer features (shown in blue) and that of the classifiers (shown in red). Logarithm of both terms converge approximately at rate $O(1/\log(t))$. (d) The dynamics of the distance between normalized centered classifier and normalized last layer feature. Logarithm of the quantity converge approximately at rate $O(1/\log(t))$ to the point of self-duality.

**Realistic Training.** We also extend our theory to realistic neural network training on benchmark dataset. To evaluate our theory, we train the VGG-13 [32] on FashionMNIST [40] without weight decay and track the convergence speed of the last layer feature to the neural collapse solution every few epochs to see how it changes during the terminal phase training. Observe that all the aforementioned quantities either decrease or stay in small values during the training process, providing implications that neural collapse can occur with sufficient training epochs.

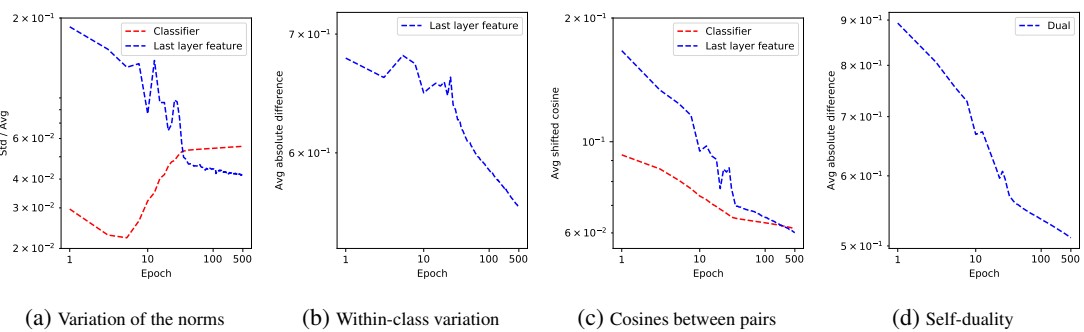

(a) Variation of the norms      (b) Within-class variation      (c) Cosines between pairs      (d) Self-duality

Figure 3: Training VGG-13 without weight decay on FashionMNIST. The $x$-axis in the figures are set to have $\log(\log(t))$ scales and the $y$-axis in the figures are set to have log scales. (a) Variation of the centered class-mean features' norms and that of the classifier's norms are below 0.1 after 500 epochs. (b) Logarithm of the within-class variation of last layer features decreases approximately linearly with respect to $\log(\log(t))$ after 100 epochs. (c) The cosines between pairs of last layer features and that of the classifiers decrease and are below 0.1 after 500 epochs. (d) The distance between normalized centered classifier and normalized last layer feature decreases during training towards self-duality.

## 5 Conclusion and Discussion

### 5.1 Conclusion

To understand the inductive bias of neural feature from gradient descent training, we build a connection between large margin inductive bias with neural collapse phenomenon and study a unconstrained

layer-peeled model in this paper. We proved that the gradient flow of the ULPM convergences to KKT point of a minimum-norm separation problem where the global optimum satisfies neural collapse conditions. Although the ULPM is nonconvex, we show that ULPM have a nice landscape where all the stationary point is a strict saddle point in the tangent space except the global neural collapse solution. Our study helps to demystify the neural collapse phenomenon, which shed light on the generalization and robustness properties during the terminal phase of training deep networks in classification problems.

## 5.2   Relationship with Other Results

Theoretical analysis of neural collapse are first provided by [24, 39, 9], they show that the neural collapse solution is the only global minimum of the simplified non-convex objective function. In particular, [39, 24] study a continuous integral form of the loss function and show that the feature learnt should be a uniform distribution on sphere. A more realistic discrete setting are studied in [9], where the constraint is on the whole feature matrix rather than individual features. All these results only relies on Jensen inequality on output logits thus can be generalized to other convex in logit losses. Our result utilize the implicit bias of the exponential like loss function to remove the feature norm constraint which is not practicable in real applications.

Though the global optimum shares good property [9], the ULPM objective is still highly non-convex. Regards optimization, [26, 29] analyze the unconstrained feature model with $\ell_2$ loss and establish convergence results to collapsed feature for gradient descent. However they fail to generalize on other more practical loss functions used in classification tasks. The analysis highly relies on the $\ell_2$ loss which turns the training dynamic to an ODE in eigenvalues.

The most relevant paper is a *concurrent* breakthrough work [43], which provide a landscape analysis about the regularized unconstrained feature model. [43] turns the feature norm constraint in [9] into feature norm regularization and still preserves the neural collapse global optimum. At the same time, [43] also show that the modified regularized objective shares a benign landscape, where all the critical points are strict saddles except the global one. Although our paper and [43] discover similar landscape results, we believe our characterization stays closer to the real algorithms used in the following two ways

- The same as [24, 39, 9], [43] only utilize the convexity in logits of the loss function. However, our analysis also explores the exponential-like property of the cross-entropy loss which will enlarge the norm of the feature. The large feature will provide better approximation to the true neural collapse problem of the normalized feature via approximating the max function via gradually scaled exponential function.
- We doesn't introduce any constraints or regularization on the feature norm, which is not applied in the realist training. Regularization on feature introduce in [43] is still different from the weight decay regularization [17]. However weight decay on homogeneous neural network is equivalent to gradient descent with scaling step size on unregularized objective [22, 41].

We summarize analysis of neural collapse in Table 1.

## 5.3   Limitation and Future Work

The convergence to neural collapse is super slow. [15] provide a loss dependent learning rate schedule and leads to $O(1/t)$ convergence rate for linear regression. It's interesting to investigate can this methodology being generalized to our setting. On the other hand, although we have shown that the ULPM have a nice landscape, we still leave the global convergence of (stochastic) gradient descent as future work for we want to provide global convergence of gradient descent combined with a plug in feature extractor.

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
