# OpenReview forum: "How Gradient Descent Separates Data with Neural Collapse: A Layer-Peeled Perspective"
_NeurIPS.cc/2021/Conference — NeurIPS 2021 Submitted_

### Official Review · Reviewer_GFas · 2021-07-15

**Rating:** 5
**Confidence:** 4

**Summary:**

Recent empirical work has identified an interesting phenomenon called Neural Collapse (NC), which says features from the trained neural network *converge* to a structural simplex, which can be viewed as the *maximal margin* solution in multi-class setting. This paper studies this phenomenon with cross-entropy loss under the assumption that the features are unconstrained and make the following contributions:
(1.) Associate the late-time gradient descent dynamics to a constraint optimization problem and relates the limit points of the parameters to KKT points of the optimization problem (thm 3.1).
(2.) The authors provides partial description about the landscape of the loss, in particular, an example showing gradient flow could fail to converge to max margin solutions (NC solutions) .

**Limitations And Societal Impact:**

Yes; the authors point out some limitation of their work;

**Main Review:**

As the multi-class analogue of the binary max margin solution, NC  has received some amount of interest recently. The goal of this paper is to study this phenomenon under cross-entropy loss with $H$, the features unconstrained. This model is a substantial simplication of the original neural network models, whose features are constrained in the sense they all rely on common parameters of the networks. Such models has been previously studied by various authors: (1) When the loss is MSE [3]  (2) cross-entropy with compact support [1] [2]. This work consider the cross-entropy setting *without* compactness assumption, in which the global minima are at infinity. Overall, this paper makes some interesting observations (e.g. Example 3.1), but the originality and significancy is not high enough to merit for publication as explained below.

(1) the unconstraint setting has been widely studies recently, including mse loss, cross-entropy + compact. Removing the comactness or normalization assumption (e.g. weight decay) does not add much extra deep insight.
(2) The techniques and framework of the paper is similar to that of [4].
(3) In my opinion, the setup "Unconstrained + Late time dynamics" is overly simple and is not of the most interesting case given existing works [1-3]. Early time dynamics and loss landscape are much more interesting and could provide more useful insights under this simple setting.

In addition, the presentation and clarity of the paper need improvement as detailed below.

[1] Neural Collapse with Cross-Entropy Loss
[2] On the emergence of simplex symmetry in the final and penultimate layers of neural network classifiers
[3] Neural collapse with unconstrained features.
[4] Gradient descent maximizes the margin of homogeneous neural networks

Some major points:

In the abstract, the authors claims that "all critical points are strict saddle points except the only global minimizers", I didn't see a theorem about this in the main text. Do you mean Thm 3.4 imply this claim? If so, please add explanation. Even this hopes, you still need the assumption $q_{min} >0$ (see below) and can not say anything in the early training phase where $q_{min} < 0$. In addition, the "$\leq $" (rather than "$<$") in eq (13)/(14) make the theorem less interesting, is it possible to strengthen it to "$<$"?

About the terminology "unconstrained layer peeled model": how is this model different than the *unconstrained model* used in e.g. [26]? Why adding the description *layer peeled* which seems unnecessary if they are the same.


The claim in the introduction "how gradient descent separates data with neural collapse" seem a bit overstate. The statement "how gradient descent separates data" is more accurate to describe the dynamics from $q_{min} <0$ to  $q_{min}  >0$, i.e. from initialization to zero classification error, rather than the terminal phase of training, which is all the NC story about.

Other comments:

Be a bit more verbal, explain the reason for $log 2$ upper bound in Thm 3.1.  Is this use to assume the classification error is zero?

"Inductive Bias of Gradient Descent" this seems non-conventional. More often than not,  people use *implicit* bias of gradient descent or *inductive* bias of architectures

Right hand side of equation (4) is missing *min*; in addition, one should use *inf* since the domain is non-compact and the minimizers are at infinity.

equation after 127, missing negative signs;

157: "collapsed margin minimize the neural collapse margin": minimize --> maximize.

170: what is $\alpha(W, H)$ do you mean $(\alpha W, \alpha H)$ ?

171: the exponent in $q_{min}$ should be 1 rather than 1/2; because 1/2 does not make the term scaling invariance;

Lemma 3.2. : the "~" above $W/H$ is not in the optimal location.   "of the
following minimum-norm separation problem": following --> above or eq (5);

In Thm C1 in appendix (same as Thm 3.4?) requires $q_{min}>0$, i don't see it in Thm 3.4 ; pleas clarify;

188: "is the angle between" --> add "cos". Suggestion: group $W$ and $H$ into one variable, say $\Theta = (W, H)$, define an inner product on this space; this should help clean up the notation a bit and avoid repeating using Trace(WW^T) + trace(HH^T) etc;

I find it helpful to bring up the (\epsilon, \delta) KKT definition to the main text;


222. "but there also exist some better points close to it thus stochastic gradient  method can easily escape form them": add reference here; form -> from

Thm 3.3 or closely related seems to already appear in previous works under the unconstraint setting; see [1-3] above;


Section 4 seems to just justify neural collapse empirical and does not provide additional information beyond what was illustrated in the original NC paper;


————————

Update:

NC is an interesting direction and the authors have obtained some very promising results. If this paper wasn’t accepted this time, I encourage the authors to polish the presentation, strengthen the results and resubmit the paper.

**Time Spent Reviewing:**

6

---

> ### Author Response · Authors · 2021-08-10
> **Official Comment**
>
> Thanks for your detailed comments and helpful suggestions. We think the typos in the paper might cause the reviewer to have some misunderstanding about our contribution. We would like to give a brief statement at first and leave a detailed response to each point later.
>
> **Regarding the contribution to the paper.** The reviewer may have missed our landscape analysis in **section 3.2 (Theorem 3.4)**. We illustrate that the ULPM with cross-entropy loss enjoys a benign global landscape on this model where all the critical points are strict saddle points except the only global minimizers which exhibit neural collapse phenomenon. This analysis closes the gap between KKT point and neural collapse solutions which is nontrivial in the non-convex setting. A detailed comparison to previous works are listed as follows:
>
> **Comparison with [1-3].** The Layer peeled model is a nonconvex model. [1-3] assumed achieved global optimum while ours analyzes the dynamic and the landscape, which is not obvious in the nonconvex setting.
>
> **Comparison with [4].** The analysis in [4] highly depends on the closed-form solution of gradient flow induced by the MSE loss and thus fails to generalize to the commonly used cross-entropy loss in classification tasks. In addition, their theoretical result relies highly on the approximation step in which they directly drop out some minor terms in the dynamics and lack rigorous analysis. Compared to theirs, our analysis covers the cross-entropy loss which is a more realistic setting in modern deep learning practice and provides rigorous analysis about the behavior of the gradient flow.
>
> **Comparison with [5].** Since the optimization problem is non-convex, [5] only guarantees convergence to a KKT point which may not be the neural collapse solution. In fact, we list a counterexample in Example 3.1. That is why we provide the second-order landscape analysis in section 3.2 to justify that except for the global optimal which can be precisely characterized by neural collapse, the other points are in fact saddle points or even not critical points which can be avoided by the stochastic gradient method. **All the proof techniques in** section 3.2  **and Appendix C are different from [5].**
>
> [1] Jianfeng Lu and Stefan Steinerberger. Neural collapse with cross-entropy loss. arXiv preprint arXiv:2012.08465, 2020.
>
> [2] Stephan Wojtowytsch and Weinan E. On the emergence of tetrahedral symmetry in the final and penultimate layers of neural network classifiers. arXiv preprint arXiv:2012.05420, 2020.
>
> [3] Cong Fang, Hangfeng He, Qi Long, and Weijie J Su. Layer-peeled model: Toward understanding well-trained deep neural networks. arXiv preprint arXiv:2101.12699, 2021.
>
> [4] Dustin G Mixon, Hans Parshall, and Jianzong Pi. Neural collapse with unconstrained features. arXiv preprint arXiv:2011.11619, 2020.
>
> [5] Kaifeng Lyu and Jian Li. Gradient descent maximizes the margin of homogeneous neural networks. arXiv preprint arXiv:1906.05890, 2019.
>
> **Response to major points:**
>
> - **Regarding the claim "all critical points are strict saddle points except the only global minimizer"**: We apologize for this typo and this "\leq" should be "<" in Theorem 3.4, which is supported by the equation between lines 246 to 247 in the appendix. the result in Theorem 3.4 is a direct result of this equation and here we indeed obtain a strict inequality. As discussed in remark 3.3 and the proof of Theorem 3.4, the proof strategy is that we first analyze those points whose gradient is not zero and obtain the inequality by the first-order condition, then we turn to those points that are not neural collapse solutions and show that the Hessian matrix has a negative eigenvalue. Then, combined with Theorem 3.3 which indicates that the global optimal solutions are neural collapse solutions, we conclude that “all critical points are strict saddle points except the only global minimizer”. The assumption “q_{min}>0” is natural when we are studying neural collapse, which is known to be a phenomenon exhibited only in the terminal phase of training.
>
> - **Regarding the name "unconstrained layer-peeled model"**: The unconstrained model in [4] is specific to MSE loss while our analysis focuses on the cross-entropy loss, to make a difference from their work and as a follow-up work of [3], we name our model as “unconstrained layer peeled model”. And a detailed comparison between [4] and our paper has been discussed in the previous response.
>
> - **Regarding the claim "how gradient descent separates data with neural collapse"**: Overall we hope to justify how the asymptotic behavior of gradient flow relates to neural collapse. Although in the terminal phase of training data has been perfectly separated, many directions can achieve such perfect classification accuracy our theoretical results show that the gradient flow will be ultimately biased to the neural collapse solutions which are symmetric and known to be better on generalization and robustness [6] Hence, we focus on this terminal stage and name the paper with “how gradient descent separate data with neural collapse”.
>
>   [6] Papyan V, Han X Y, Donoho D L. Prevalence of neural collapse during the terminal phase of deep learning training[J]. Proceedings of the National Academy of Sciences, 2020, 117(40): 24652-24663.
>
> **Response to other comments:**
>
> - **Regarding the $\log 2$​ upper bound in Theorem 3.1**: Yes, the assumption is here to guarantee all features have been separated perfectly. The proof has been provided in lines 64-66 in the appendix.
> - **Regarding the meaning of $\alpha(W,H)$​**: It means $(\alpha W,\alpha H)$​ actually.
> - **Regarding the comment "In Thm C1 in appendix (same as Thm 3.4?) requires qmin>0, i don't see it in Thm 3.4 ; pleas clarify;"**: Sorry, it is actually a typo and we do need this requirement since all of our discussion is under the terminal phase of training.
> - **Regarding the reference for escaping saddle points**: We would like to refer to [7] [8] for a discussion about how gradient-based optimization methods avoid saddle points. There are two viewpoints for this fact: the first is that for those saddle points, gradient flow will not converge along the direction in the eigenspace of negative eigenvalue, so with probability 1 the initialization has a non-zero projection in such space and will not converge to those saddle points [7]; the second is that if we use stochastic gradient method, with probability 1 that the random noise has a non-zero projection in the eigenspace of negative eigenvalue and the optimization algorithm can escape the saddle points in polynomial time [8].
>   [7]Lee J D, Simchowitz M, Jordan M I, et al. Gradient descent converges to minimizers[J]. arXiv preprint arXiv:1602.04915, 2016.
>   [8]Ge R, Huang F, Jin C, et al. Escaping from saddle points—online stochastic gradient for tensor decomposition[C]//Conference on learning theory. PMLR, 2015: 797-842.
> - **Regarding the comment "Thm 3.3 or closely related seems to already appear in previous works under the unconstraint setting; see [1-3] above;"**: Yes, but Theorem 3.3 is not our main contribution. The main result is Theorem 3.4 which characterizes that the neural collapse solutions are the only minimizer in the sense that the other critical points will be strict saddle points, which are not covered in [1-3]. For the consideration of completeness, we still provide this theorem here.
> - **Regarding the comment "Section 4 seems to just justify neural collapse empirical and does not provide additional information beyond what was illustrated in the original NC paper;"**: Although neural collapse with weight decay has been intensively investigated in previous works, it remains to be unexplored that if weight decay is necessary for the emergence of neural collapse. Predicted by our theoretical analysis, we provide empirical results to justify that the emergence of neural collapse solutions should attribute to the property of optimization algorithm and loss function, the implicit regularization itself is sufficient to lead to neural collapse solutions. Moreover, it's observed that in our setting the convergence rate is approximate $O(1/\log(t))$ and it's interesting to further explore the convergence rate of neural collapse in different settings.
> - We would like to thank the reviewer for the kind suggestions on the presentation and clarity of our paper and we must apologize for much unnecessary confusion. We also find that the submission version of our paper contains many typos and some results are not well presented. Indeed, after our paper was submitted to NeurIPS, we continued to make extensive efforts to polish the writing and improve the presentation. A polished version has been made and we assure you’d see a significantly improved version of our paper, provided that our paper is accepted.
>
> We hope our answer can clarify the confusions and the reviewer will be satisfied with our response, and convinced to increase the score, thus helping this paper get accepted.

---

> > ### Comment · Reviewer_GFas · 2021-08-18
> > **Update**
> >
> > Thanks for the detailed response.
> >
> > In light of the strict inequality `<` ( v.s. `\leq`), I increase the score to 5, since the `\leq` is not very useful.
> >
> > However, I still don't think the paper meet the bar of acceptance. My following perspective does not change:
> > >(3) In my opinion, the setup "Unconstrained + Late time dynamics" is overly simple and is not of the most interesting case given existing works [1-3]. Early time dynamics and loss landscape are much more interesting and could provide more useful insights under this simple setting.
> >
> > Note that I mean `Early time dynamics and **early time** loss landscape above.
> >
> > Alternatively, the authors may consider the positively homogeneous networks (e.g. Relu) and prove the convergence to NC-like minima in the late time dynamics (e.g. Kaifeng Lyu and Jian Li) and perhaps even study the direction convergence like in [1]. Either result will strengthen the contribution.
> >
> > [1] Ziwei Ji, Matus Telgarsky. Directional convergence and alignment in deep learning

---

> > > ### Author Response · Authors · 2021-08-19
> > > **Response to the update**
> > >
> > > Thanks for increasing the score and your suggestions. The condition $q_{min}(W, H)>0$​​ is here to provide an analysis of how gradient flow converges to the first order KKT point. Actually, the landscape analysis can be generalized to the early time setting easily. To see this, please check out the proof of Theorem C.2. in the appendix. The condition $q_{min}>0$​​ is used to guarantee $\lambda<0$ in lines 229-230. However, when $\lambda>0$, by equation (73) we know $\frac{d^2}{dt^2}h(\phi(t))=0$, you can change the minus sign in equation (75) to be a plus sign. Then just change the $-\lambda$ to be $\lambda$ in equations between lines 242-243, lines 246-247 and in line 243, all of our analysis holds still. The remaining case is $\lambda = 0$. When $\lambda = 0$, we even do not need the analysis between lines 240-243, equations between lines 246-247 only are enough to find a decreasing direction since $\nabla L(WH)$ is of rank $K-1$​​​. Since we aim to illustrate how neural collapse emerges in the terminal phase of training, we didn't state our theorem under the most general conditions in the submitted version and we will ensure it to be included in our new version.
> > >
> > > In short, to determine which solution will be ultimately found by gradient flow with stochastic perturbation (a corollary of the landscape analysis), we don't need the terminal training phase assumption. But with the help of assumption $q_{min}(W, H)>0$, we can have a more refined analysis about how gradient flow converges in direction and obtain an exact convergence rate (the proof is similar to [1] it will be included in our next version). How to generalize this convergence analysis to early time dynamics is an interesting direction and we will continue to work on this, but currently, this assumption is necessary for all of the implicit bias analysis in the nonlinear setting [1] [2] [3]. We'll make the assumptions and the conclusions of the two parts in our paper more clearly in the next version.
> > >
> > > We must apologize for bringing this unnecessary confusion (we merged the assumptions of the two-section in the first version and leads to the unnecessary assumption for the landscape analysis) and we hope our answer can improve your evaluation of our paper.
> > >
> > > [1]Kaifeng Lyu and Jian Li. Gradient descent maximizes the margin of homogeneous neural networks. arXiv preprint arXiv:1906.05890, 2019.
> > >
> > > [2]Ji, Ziwei, and Matus Telgarsky. "Directional convergence and alignment in deep learning." *arXiv preprint arXiv:2006.06657* (2020).
> > >
> > > [3]Nacson, Mor Shpigel, et al. "Lexicographic and depth-sensitive margins in homogeneous and non-homogeneous deep models." *International Conference on Machine Learning*. PMLR, 2019.

---

> > > ### Author Response · Authors · 2021-08-21
> > > **Looking forward to the response**
> > >
> > > We wonder if our answer addressed your concern about the early time landscape. If the reviewer still has questions, we’d love to have a further explanation. If the concerns are addressed, we hope the reviewer can improve the evaluation of our paper.

---

> > > > ### Comment · Reviewer_GFas · 2021-08-25
> > > > **final update**
> > > >
> > > > Thanks for the update and I am glad that you improve your results to include loss landspace + early time.
> > > >
> > > > The core problem in this setting is *HOW* does gradient descent/flow separate the data (q_min < 0 --> q_min > 0). There seems to be no existing paper that answers this question (to the best of my knowledge). I encourage the authors to tackle this problem which could be a very novel contribution to the field. In addition, the presentation of the paper is not in high quality as mentioned by other reviewers. This makes it hard for AC/reviewers to argue in favor of acceptance *unless* the contribution is strong enough to off-set its weakness.
> > > >
> > > > In sum, I am not convinced to raise the score again.

---

> > > > > ### Author Response · Authors · 2021-09-06
> > > > > **Waiting to hear from the reviewer**
> > > > >
> > > > > Dear reviewer:
> > > > > In our most recent response, we think that we have addressed your concerns. However we still not heard back from you as to whether you feel we have addressed your concerns. We are willing to discuss if the reviewers have further concerns but it seems that the communication is blocked. Whether you update your score may be critical to our paper's acceptance. The decision deadline is coming, we are here still waiting for the response from the reviewer.
> > > > >
> > > > > For convenience, we paste our response here again.
> > > > >
> > > > > We think that the reviewer may underestimate the novelty of removing the terminal training stage assumption for the landscape results. Actually, the landscape results in early time training have already answered the question of HOW does gradient descent/flow separates the data. Combining the landscape results with [1,2] we can know that the (stochastic version) gradient descent converges to the neural collapse solution with probability 1. On the other hand, the assumption contributes to a detailed analysis of the learning dynamics, for example, the monotonicity of the margin (which is not true in the early time) and the log(1/t) convergence rate (escaping saddle point can be hard provably).
> > > > >
> > > > > The landscape result is considered as our main contribution and has answered the question “ HOW does gradient descent/flow separate the data”. There is no existing paper that answers the convergence results including the early-time dynamics, all of the current analysis needs the terminal phase condition [3,4,5]. In this paper, however, we adopt the landscape analysis which helps us to remove the terminal phase condition and conclude a convergence result. As the reviewer said, it's a very novel contribution to the field and we believe the contribution can offset the weakness in writing.
> > > > >
> > > > > We must apologize for the confusion caused by the linguistic errors. But we believe that they do not hide the proof ideas and impair contributions of our work. At the same time, Reviewer XiQd thinks our paper is “easy to follow”，Reviewer HKcz agrees that “our proof technique is clear”, thus we believe that our contribution and theoretical results are demonstrated clearly. Under this situation, we hope that this paper can be reevaluated. We don’t want to find our paper being rejected just because of linguistic errors and we are willing to address any further concerns related to our paper.
> > > > >
> > > > > [1] Lee J D, Simchowitz M, Jordan M I, et al. Gradient descent only converges to minimizers[C]//Conference on learning theory. PMLR, 2016: 1246-1257.
> > > > >
> > > > > [2] Ge R, Huang F, Jin C, et al. Escaping from saddle points—online stochastic gradient for tensor decomposition[C]//Conference on learning theory. PMLR, 2015: 797-842.
> > > > >
> > > > > [3]Kaifeng Lyu and Jian Li. Gradient descent maximizes the margin of homogeneous neural networks. arXiv preprint arXiv:1906.05890, 2019.
> > > > >
> > > > > [4]Ji, Ziwei, and Matus Telgarsky. "Directional convergence and alignment in deep learning." arXiv preprint arXiv:2006.06657 (2020).
> > > > >
> > > > > [5] Nacson, Mor Shpigel, et al. "Lexicographic and depth-sensitive margins in homogeneous and non-homogeneous deep models." International Conference on Machine Learning. PMLR, 2019.

---

> > > > > > ### Comment · Area_Chair_CNxP · 2021-09-06
> > > > > > **Thank you (area chair here).**
> > > > > >
> > > > > > Dear Authors,
> > > > > >
> > > > > > Thank you for sharing your concerns and detailed comments.  Please rest assured that detailed discussions continued internally.
> > > > > >
> > > > > > Thank you for your time!

---

### Official Review · Reviewer_XiQd · 2021-07-16

**Rating:** 7
**Confidence:** 4

**Summary:**

The work studied the inductive bias of the last-layer features and classifiers from neural networks with cross-entropy loss, based on the unconstrained layer-peeled model. The work (i) builds a relationship between the max-margin analysis with the neural collapse and provides the inductive bias analysis to the unconstrained feature, and (ii) shows that the loss function without weight decay enjoys a benign landscape and characterize the neural collapse property of the global minimizer.

**Limitations And Societal Impact:**

The result is based on a simple layer-peeled model. It could be interesting how to extend the analysis to study deep networks under more practical settings.

The current study only focuses on the training. It would be interesting how to study generalization and robustness based upon the current analysis.


**Main Review:**

The work provides an interesting study on the theoretical investigation of the neural collapse phenomenon of the last-layer features and classifiers of deep networks based on the layer-peeled model, where the neural collapse has been recently discovered empirically [28]. The presentation of the work is clear and easy to follow. However, the reviewer finds that the two results on max-margin analysis and landscape analysis are a bit disconnected. Below are some more detailed comments:

* Section 3.1: In Theorem 3.1, can the assumption L_CE(t_0) < log 2 be achieved using some initializations? How stringent is this assumption for the result to hold? Some discussion on explaining this could be helpful here.

Also, for the results in Section 3.1, it would be great if one can show that the global solution of (2.3) in the tangent space (or limiting point of the flow of (2.3)) is the neural collapse and KKT solution for 3.1, it could make the story more coherent to Section 3.2. Currently, it seems that the two sections are a bit disconnected.

* Section 3.2, the reviewer finds that the claim about loss could be a bit misleading. Although the loss does not have any weight decay, nor constraints at first glance, the analysis is conducted over the tangent space of a manifold ||W||_F^2 +||H||_F^2 = C, which can be reviewed as a constraint on the loss function.

Also, for Theorem. 3.4, it would be better to show a second-order directional descent instead of properties on loss function (zero-order). In the current form of (13), the direction might not strictly decrease the function value.


**Time Spent Reviewing:**

6 hours

---

> ### Author Response · Authors · 2021-08-10
> **Official Comment**
>
> Thanks for your insightful comments and kind suggestions. Please find a response to each point below:
>
> - **Regarding the assumption** $\mathcal{L}_{CE}(t_0)<\log 2$: The assumption is here to guarantee all features have been separated perfectly.  You can find proof in lines 64-66 in the appendix and its validity can be justified since neural collapse is found only in the terminal phase of training in the deep neural network, where the training accuracy has achieved 100%. To achieve this, consider a simple case when each class has only one data point and the data point is arranged in the class order (i.e., the $i$-th column in $\boldsymbol{H}$ belongs to the $i$-th class ), we only need to initialize the $\boldsymbol{W, H}$ to be both identities. In the general case, we can just copy the initialization for each data within the similar class and do some appropriate permutation to adjust to a general class order in $\boldsymbol{H}$​.
>
> - **Regarding the connection between section 3.1 and 3.2**: Please refer to Theorem 3.2 and Theorem 3.3, we show that the global solution and the limit point of flow are both neural collapse solutions. And as discussed in the first paragraph in section 3.2, in section 3.1 we have shown that the gradient flow will converge to a KKT point of a mini-norm separation problem and its global solution is a neural collapse solution. However, due to its non-convexity, there still exists a gap between KKT point and neural collapse which has been verified by constructing a counterexample in Example 3.1. Then we provide analysis in section 3.2 to show that although there exist some KKT points that are not neural collapse solutions, they are in fact saddle points in the tangent space and thus can be avoided by the gradient descent or stochastic gradient descent [1] [2].
>
>   [1]Lee J D, Simchowitz M, Jordan M I, et al. Gradient descent converges to minimizers[J]. arXiv preprint arXiv:1602.04915, 2016.
>
>   [2]Ge R, Huang F, Jin C, et al. Escaping from saddle points—online stochastic gradient for tensor decomposition[C]//Conference on learning theory. PMLR, 2015: 797-842.
>
> - **Regarding the comment on the claim about the loss in section 3.2**: Here we write (3.1) as a constraint problem, but the constraint is introduced by the implicit regularization effect of gradient flow on our ULPM objective (2.3). Since the training dynamics will diverge to infinity, we hope to justify the diverge direction is highly related to neural collapse and an appropriate normalization is needed, that’s why it appears to be a constraint optimization form. Our goal is to justify that the neural collapse phenomenon is caused by the properties of the loss function and training dynamics rather than an explicit regularization or constraint, which seems to be necessary for previous works.
>
> - **Regarding the suggestion of showing second-order directional descent**: Sorry for this typo and this "$\leq$​" should be "<", you can check the equation between lines 246 to 247 in the appendix, the result in Theorem 3.4 is a direct result of this equation and here we indeed obtain a strict inequality. We do recognize that showing a second-order directional descent may help the reader to better understand why it is a saddle point, thanks for your suggestions!
>
> - We also find that the submitted version contains many typos and is not well presented. We have made a more polished version after the deadline and added additional results and discussion including:
>
>   - New experiment results justify that in our setting, neural networks can achieve comparable performance to the commonly used weight decay setting.
>
>   - Comparison of the convergence speed to neural collapse between different settings. In particular, we find that the training with weight decay converges much faster than training without weight decay. But once appropriate normalization techniques are used (such as increasing the learning rate to adjust to the parameter norm), training without weight decay can achieve comparable convergence speed to training with weight decay.
>
>   - We add more discussion about the proof intuition and connection between different theorems.
>
>     Unfortunately, we can not submit the current version immediately due to the policy, but we assure you will see a significantly improved version of our paper once it is accepted.
>
>   Unfortunately, we can not submit the current version immediately due to the policy, but we assure you will see a significantly improved version of our paper, provided that our paper is accepted.

---

> ### Author Response · Authors · 2021-08-22
> **An update on response.**
>
> After discussion with reviewer Dsaf, we discovered that we can remove the terminal stage assumption $q_{min}(W, H)>0$​ (i.e. $\log(\mathcal{L})<\log 2$)​ for landscape analysis. The condition $q_{min}(W, H)>0$​​ is here to provide an analysis of how gradient flow converges to the first order KKT point. Actually, the landscape analysis can be generalized to the early time setting easily. To see this, please check out the proof of Theorem C.2. in the appendix. The condition $q_{min}>0$​​ is used to guarantee $\lambda<0$ in lines 229-230. However, when $\lambda>0$, by equation (73) we know $\frac{d^2}{dt^2}h(\phi(t))=0$, you can change the minus sign in equation (75) to be a plus sign. Then just change the $-\lambda$ to be $\lambda$ in equations between lines 242-243, lines 246-247 and in line 243, all of our analysis holds still. The remaining case is $\lambda = 0$. When $\lambda = 0$, we even do not need the analysis between lines 240-243, equations between lines 246-247 only are enough to find a decreasing direction since $\nabla L(WH)$ is of rank $K-1$​​​. Since we aim to illustrate how neural collapse emerges in the terminal phase of training, we didn't state our theorem under the most general conditions in the submitted version and we will ensure it to be included in our new version.
>
> In short, to determine which solution will be ultimately found by gradient flow with stochastic perturbation (a corollary of the landscape analysis), we don't need the terminal training phase assumption. But with the help of assumption $q_{min}(W, H)>0$, we can have a more refined analysis about how gradient flow converges in direction and obtain an exact convergence rate (the proof is similar to [1] it will be included in our next version). How to generalize this convergence analysis to early time dynamics is an interesting direction and we will continue to work on this, but currently, this assumption is necessary for all of the implicit bias analysis in the nonlinear setting [1] [2] [3]. We'll make the assumptions and the conclusions of the two parts in our paper more clearly in the next version.
>
> We must apologize for bringing this unnecessary confusion (we merged the assumptions of the two-section in the first version and leads to the unnecessary assumption for the landscape analysis) and we hope our answer can improve your evaluation of our paper.
>
> [1]Kaifeng Lyu and Jian Li. Gradient descent maximizes the margin of homogeneous neural networks. arXiv preprint arXiv:1906.05890, 2019.
>
> [2]Ji, Ziwei, and Matus Telgarsky. "Directional convergence and alignment in deep learning." *arXiv preprint arXiv:2006.06657* (2020).
>
> [3]Nacson, Mor Shpigel, et al. "Lexicographic and depth-sensitive margins in homogeneous and non-homogeneous deep models." *International Conference on Machine Learning*. PMLR, 2019.

---

### Official Review · Reviewer_HkCz · 2021-07-16

**Rating:** 6
**Confidence:** 4

**Summary:**

This paper studies the optimization dynamics of the layered peeled model - a simplified model for the study of neural collapse in overparametrized deep learning, wherein the last layer features are regarded as free optimization variables. This paper studies this problem in an unconstrained setting; that is, without using weight decay or norm constraints. To this end, the authors tie their problem to that of a maximum margin (min norm) alternative, and show that the former converges (in direction) to the latter, even if their magnitudes diverge. By studying the max margin problem, the authors then show that no strict saddles exist, and thus noisy-gradient descent type algorithms can converge to a global minimum of the problem. Finally, the authors demonstrate this behavior numerically in a toy problem as well as on the Fashion MNIST dataset.

**Limitations And Societal Impact:**

Adequately addressed.

**Main Review:**

This paper is interesting, and it contributes to demystifying the neural collapse phenomenon. The sequence of ideas and proof technique is mostly clear. I have the following observations:

Main comments:
* When comparing to similar recent contributions, the authors stress that they study the unconstrained version of the layered peeled model and argue that this is more realistic and reflective of the current practice on training deep learning algorithms. This reviewer is somewhat confused by this, as the experimental results in [29] (which first demonstrated neural collapse) do use weight decay. Moreover, the results here regard convergence in direction, which is typical of the analysis of gradient descent for logistic loss. Naturally, this involves the weights norms diverging. This also separates from what is observed in practical settings, where the weights do not diverge. As a result, I feel like the comment on their setting being "closer to current practice" is a bit overstated. The analysis and proof technique, and the assumptions and settings, is simply different.

* The authors present results for convergence of gradient flow - that is, assuming continuous dynamics. Can the authors comment on the extensions of these ideas to discrete time (stochastic) gradient descent?

* Respectfully, this paper is hard to read at times simply because of the huge number of typos and grammatical mistakes. A non-exhaustive list follows below.

* On their experimental results section, the authors first demonstrate the training dynamics on their ULPM, and then on the Fashion MNIST dataset. In the latter case, the variation of the classifiers norm increases rather decreases, which basically contradicts the analysis presented here. Naturally, there are differences between the practical setting and the one analyzed theoretically, but can the authors expand on the reasons for this behavior?

* On theorem 3.4, the authors show that there exist directions in the tangent space that lead to lower value of the objective function. I'm confused by the inequality on Eq. (13), which is not a strict inequality. As a result, this does not imply necessary a descent direction, as I think is needed for the argument - though I might be missing or misunderstanding something. Could the authors clarify this point?

Non-exhaustive minor comments:
- L53, "use it provide" -> use it to provide
- The sentence on line 77 "To understand how gradient or its variants descent helps deep learning to find solutions with good generalization performance on the test set." has no subject, and no meaning.
- L85: "that critical points other than the good ones always lies" -> critical point lie.
- L98 "within in the k-th class" -> within the k-th class
- L103: "to be simplex equiangular" -> to be a simplex..
- NC2: closing parenthesis missing
- L107: "satisfies neural collapse conditions or is neural collapse solution" -> satisfies the neural collapse conditions or is a neural collapse solution.
- L114: "Similar simplification is common used [..] but ours don’t" -> but ours doesn't.
- Gradient flow is never defined. Instead, the authors just define the gradients, on line 127. I.e, the definition on Eq (14) in the appendix should be brought to the main text.
- Line 157 "collapsed margin minimize" -> minimizes
- L202: "gradient flow move" -> moves
-L 212 "simple gradient descent get stuck" -> gets stuck
- L231 "Now we turns" -> Now we turn
- Extra parenthesis on Eq 12
- "Our next result justify" -> justifies
- "for every suboptimal points, there exist a direction in the tangent space such that move along this direction will leads to a lower objective
value." -> for every suboptimal point, there exists a direction in the tangent space such that moving along this direction will lead to a lower objective value
- Many punctuation marks missing or incorrect (Eq 13, 14, 12, 10, 9, 8)
- "For a large family of stochastic optimization algorithm" -> algorithms
- "we observe that logarithm" -> that the logarithm
- "We doesn’t introduce" -> we don't introduce.
Again, this list is not exhaustive.

**Time Spent Reviewing:**

2.5

---

> ### Author Response · Authors · 2021-08-10
> **Official Comment**
>
> Thanks for your insightful comments and careful correction. Please find a response to each point below:
>
> - **Regarding the comment "When comparing to similar recent contributions......":** Although weight decay is widely used in practice, it's still different from direct constraint or regularization on features due to the high nonlinearity of neural networks. Here we hope to justify that our setting will be closer to the realistic setting since the feature regularization or constraint is never used in the realistic training, but neural networks can still achieve comparable performance even in the absence of explicit regularization (i.e., weight decay). We would like to refer to [1] as an empirical justification. It is observed that neural networks continue to perform well after all the regularizers are removed. See section 3 in [1] for detailed results and discussion. Except for practical consideration, it’s hard to plug in a neural network when features are constrained while ours appears to be more capable for further analysis on real neural networks with the features gaining more freedom. By removing such explicit regularization and constraint, it’s hoped that more theoretical results on the real neural networks can be inspired.
>
>   [1] Chiyuan Zhang, Samy Bengio, Moritz Hardt, Benjamin Recht, and Oriol Vinyals. Understanding deep learning requires rethinking generalization. arXiv preprint arXiv:1611.03530, 2016.
>
> - **Regarding the comment on the extension to discrete-time gradient descent**: Due to the limitation of space, we only provide the results for continuous-time dynamics to illustrate the essential part of the dynamics, but it’s promising that we can use some techniques similar to [2] to finish the proof in the discrete-time cases. However, for the stochastic gradient descent, the analysis may be more complicated since the parameter will go to infinity, we need to carefully adjust the noise term or apply some normalization techniques. Recent work [3] has obtained some convergence results in the linear model and it is interesting to explore how to generalize their techniques to a nonlinear model and provide insights for neural collapse.
>
>   [2]Kaifeng Lyu and Jian Li. Gradient descent maximizes the margin of homogeneous neural networks. arXiv preprint arXiv:1906.05890, 2019.
>
>   [3]Nacson M S, Srebro N, Soudry D. Stochastic gradient descent on separable data: Exact convergence with a fixed learning rate[C]//The 22nd International Conference on Artificial Intelligence and Statistics. PMLR, 2019: 3051-3059.
>
> - **Regarding the experiment**: In practice, the convergence behavior might be affected by additional tricks (e.g., batch normalization) and the architecture. We would like to refer to Fig.2 in the original neural collapse paper [4], where you can find that the curve about the variation of classifiers norm also increases or stays the same for a long time. But the variation will always stay at a lower value (5*10-2) and we think that it is caused by the slight difference between the realistic neural networks and the simplified model.
>
>   [4] Vardan Papyan, XY Han, and David L Donoho. Prevalence of neural collapse during the terminal phase of deep learning training. Proceedings of the National Academy of Sciences, 117(40):24652–24663, 2020.
>
> - **Regarding Theorem 3.4**: Sorry for this typo and this "$\leq$" should be "<", you can check the equation between lines 246 to 247 in the appendix, the result in Theorem 3.4 is a direct result of this equation and here we indeed obtain a strict inequality.
>
> - **Regarding the typos and grammatical mistakes**: Thanks again for your careful correction. We also find that the submission version of our paper contains many typos and some results are not well presented and we must apologize for much unnecessary confusion we have brought. Indeed, after our paper was submitted to NeurIPS, we continued to make extensive efforts to polish the writing, improve the presentation, and added additional results and discussion including:
>
>   - New experiment results justify that in our setting, neural networks can achieve comparable performance to the commonly used weight decay setting.
>
>   - Comparison of the convergence speed to neural collapse between different settings. In particular, we find that the training with weight decay converges much faster than training without weight decay. But once appropriate normalization techniques are used (such as increasing the learning rate to adjust to the parameter norm), training without weight decay can achieve comparable convergence speed to training with weight decay.
>
>   - We add more discussion about the proof intuition and connection between different theorems.
>
>     Unfortunately, we can not submit the current version immediately due to the policy, but we assure you will see a significantly improved version of our paper once it is accepted.

---

### Official Review · Reviewer_6dN9 · 2021-07-17

**Rating:** 4
**Confidence:** 3

**Summary:**

This paper studies representations of neural networks trained with cross entropy, when trained beyond zero training error, utilizing the unconstrained layer peeled model (ULPM). ULPM is motivated by the fact that many neural networks are highly overparametrized and assumes that the representations, on which the last linear layer acts, can be optimized freely, i.e.  they are unconstrained.
In this model, the authors derive several theoretical results, related to the recently observed neural collapse [28]. In particular, they derive a relation to max-margin analysis, show that neural networks converge to Karush–Kuhn–Tucker (KKT) points of a certain minimum-norm separation problem, and show that the loss landscape is benign, as it has no spurious minima.

**Limitations And Societal Impact:**

The authors addressed some limitations. These are however not the limitations I see in this work, which are rooted in the unrealistic setting that models are trained without weight decay.

**Main Review:**

In its current form, the paper is not ready for publication. This is mostly due to clarity issues. Aside from linguistic errors and typos, there are whole paragraphs which are very difficult to follow. As a consequence, also the theoretical contribution suffers, as its meaning and significance cannot be assessed adequately. I believe however, that this paper would make a decent contribution, once the major issues are resolved.

### Strengths:

- The paper belongs to a group of recent works which study the so called neural collapse in the unconstrained layer peeled model. However, in contrast to previous work, the authors do not only study the *global* minimum, but also *local* optimization dynamics and curvature.
- The paper does a good job positioning itself within the related work in section 5.2
- The presented example 3.1 was very welcome. It proves the existence of problematic points in the loss landscape, which is not obvious, and motivates the subsequent theory. I would have liked it even more, if it was paired with a figure.

### Weaknesses:

- As already mentioned, the paper is hard to follow.
  - Paragraphs, which are quite difficult to understand include remark 3.1, the discussion of reference [43] between lines 300 and 309, and the first paragraph of section 3.2, which contradicts itself.
  - line 32 to 34: The terms "within-class cross-sample features" and "interpolating the in-sample training data" need more background.
  - line 84: I don't under stand what is meant with "balanced superpositions of symmetric copies of the ground truth according to the hidden symmetries in the objective function".
  - In theorem 3.1, it is not clear, what is precisely meant with "along the direction of an Karush-Kuhn-Tucker (KKT) point". Does this mean, that the referred to limit point and KKT point are positive multiples of each other?
  - It seems, that a combination of theorem 3.1 and 3.2 would be the main point of the 3.1.2. Convergence Results section. However, this combination is not discussed at all.
  - If Lemmas 3.2 and 3.3 should be kept in the main text, then these need to be discussed in more detail.
  - Definition 3.2 defines the tangent space at a point as the point’s orthogonal complement. This indicates that this point lies on a sphere. However, this does not seem to be the case.
  - What is meant with "collapsed margin minimize the neural collapse margin" (line 157). Similarly, it is unintuitive, why the quantity in definition 3.1 is called the neural collapse margin, as, a priori, it is not connected to neural collapse. As a reader, I expected the neural collapse margin to be defined via a minimum over all $W$ and $H$.
  - It is unclear, whether there is a difference between the Layer-Peeled Model (1 Introduction) and the Unconstrained Layer-Peeled Model (2.2 Problem Setup)
  - I will list some minor linguistic errors at the end of the review, so that the authors can correct them.

- There are several claims, which need to be discussed in more detail, or appear to be wrong
  - In line 57, it is claimed that "The analysis provides insights on how gradient descent separates data during the training of neural networks with neural collapse and the benefit of training after interpolation on generalization and robustness". However, robustness and generalization aspects appear not to be covered in this work.
  - Caption of table 1: "We provide strongest theoretical results with minimum modification on the training objective function" (compared to the related work). As already mentioned, I was unable to understand the authors argument, why their work provides stronger results than [43]. Still, this is a bold claim.
  - line 62: "Our result doesn’t introduce any extra feature norm constraint or feature norm regularization, which are not commonly used in the realistic deep learning."\
  While this is true, one might argue that the omission of weight decay, as done in the analysis, is also "not commonly used in the realistic deep learning". Similarly, line 116 ("Similar simplification is common used in previous theoretical works [24,9,39,43], but ours don’t have any constraint or regularization on features and stands closer to realistic neural network models.")  needs more justification.
-  line 92: "For simplicity, we assume the dataset is balanced". It seems that neural collapse only happens in the balanced situation. Therefore this assumption is not made due to simplicity.

- The experiments are rather limited. The authors present two experiments. The first one is a simulation in the UPLM regime, which confirms the theoretical findings. However, considering, that these have already been proven in this work, the results of this experiment are unsurprising. I would advise the authors to move this to the supp. mat. and use the freed space for studying the realistic case in more detail.\
The second experiment, i.e., the "realistic setting", investigates neural collapse on a VGG-13 net. Considering that such experiments were already performed by [28] (however, with weight decay), I do not see much benefit in this experiment in its current form.
Furthermore, it seems like the experiments consist of only a single run, so the statistical robustness of the findings cannot be assessed.

- Throughout the paper, it is assumed, that training neural networks without weight decay is a realistic setting. This needs justification, either by a reference or through experiments. It would also be interesting to see, whether the speed of collapse differs for models trained with and without weight decay.


**Typos etc:**

There are many typos and similar errors in the paper. In the following, I will report ones I found only on page 7.

- line 215: exists -> exist
- line 216: direction -> directions
- line 221: start a new sentence between "it" and "thus".
- line 222: form -> from
- line 228: conditions- > condition
- line 231: turns -> turn
- line 235: justify -> justifies, exist -> exists
- line 236: move -> moving, leads -> lead
- line 240: there is a dot at the start of the line. Further more -> Furthermore
- line 245: algorithm -> algorithms
...

**References** \
as in the paper.

**Time Spent Reviewing:**

8-10

---

> ### Author Response · Authors · 2021-08-10
> **Official Comment**
>
> Thanks for your insightful comments and suggestions! We also find that the submission version of our paper contains many typos and some results are not well presented. Indeed, after our paper was submitted to NeurIPS, we continued to make extensive efforts to polish the writing and improve the presentation. The version in our hands is much more readable, which however could not be uploaded to OpenRiview at the moment because of the policy. Here we’d like to apologize for much unnecessary confusion and assure you’d see a significantly improved version of our paper, provided that our paper is accepted.
>
> Before starting response to each point, we would like to give **a summary about our logic** first to help the reviewer have a better understanding: In section 3.1 we have shown that the gradient flow can find KKT points of the mini-norm separation problem (3.1) in Theorem 3.1 and its global optimal solution is the neural collapse solution in Theorem 3.2. However, given that (3.1) is non-convex, there still exists a large gap between KKT points and globally optimal solutions, and we have found a counterexample in Example 3.1. That motivates us to study the second-order landscape in section 3.2 and show that except for the only global minimizer, the other KKT points are in fact saddle points and this result closes the gap between KKT points and neural collapse.
>
> The detailed response to each point is as follows:
>
> - **Regarding the paragraphs that the reviewer finds them hard to follow:**
>
>   - **Remark 3.1:** Remark 3.1 aims to justify that the mini-norm separation problem is equivalent to a max neural collapse margin problem. It describes the same proof process as shown in appendix Theorem B.2. We’ll conclude a more formal proof in the next version.
>
>   - **The discussion of reference [43] between lines 300 and 309**: Here we aim to emphasize the difference of our analysis to theirs in two aspects: (1) Their analysis only needs the Jensen inequality to deal with loss function, thus can apply for a large family of convex functions. However, our analysis in section 3.1 illustrates the uniqueness of cross-entropy that the implicit regularization of gradient flow is sufficient to find a neural collapse solution even in the absence of explicit regularization or constraint. (2) They remove the constraints on feature norm but add a regularization on feature norm instead, which is never used in practice and limited in extension . In comparison, we consider a scenario where the features are completely free of constraints. which can admit a further exploration by using a real neural network to replace the feature.
>
>   - **The first paragraph of section 3.2** As discussed above, here we aim to show that the global optimality of neural collapse solution is not enough to guarantee gradient flow to converge to it since the objective is nonconvex.
>
>   - **Meaning of the terms "within-class cross-sample features" and "interpolating the in-sample training data"**: The first term refers to the features that belong to the same class but different data points. The second term refers to the fact that the classifier has perfectly separated all the training data points.
>
>   - **Line 84, the meaning of "balanced superpositions..."**: Generally speaking, the landscape of the non-convex function may have many minimums and hard to analyze. However, recent works show that for a large family of non-convex functions, the landscape is highly symmetric and those minimums are just copies of each other. Essentially the minimum may be unique if we ignore its symmetric copies. We would like to refer to [1] as a survey of previous works. In our case, if you multiply an orthogonal matrix on W and its transpose on H, the output will not change and our analysis in section 3.2 shows that the neural collapse solutions are the unique minimizer.
>
>     [1] From symmetry to geometry: Tractable nonconvex problems
>
>   - **The meaning of "along the direction of a KKT point"**: Yes, it means they are parallel.
>
>   - **The combination of Theorem 3.1 and 3.2**: As discussed in lines 174-180, Theorem 3.1 illustrates that the convergence direction is parallel to a KKT point of (5). To build the relationship between (5) and neural collapse, we then provide Theorem 3.2 to show that the global optimal solution of (5) is a neural collapse solution. So the remaining work is to close the gap between the KKT point and global optimal solution which has been done in section 3.2.
>
>   - **Detailed discussion about Lemma 3.2 and 3.3:** Here we want to provide some intuitions on why gradient flow can find the KKT point. Some basic idea is that through straightforward computation the violation of KKT conditions can be bounded by the margin and the angle between parameter and gradient. So the remaining work is to investigate their asymptotic behavior.
>
>   - **The definition of tangent space**: Since the ULPM objective will always decrease along the direction of the current point and the optimum is attained only in infinity, as discussed in section 3.1, we consider directional convergence. In this case, we consider the optimality on a sphere. By Taylor’s expansion, we only need to analyze the gradient and Hessian in such tangent space to obtain the optimality on the sphere. This fact has been included and proven rigorously in the appendix and we would like to refer to lines 229 to 236 in the appendix for a formal discussion. The idea is that the sphere is a smooth manifold that can be locally approximated by the tangent space at each point.
>
>   - **The meaning of neural collapse margin**: Sorry for the confusion, we aim to express that the neural collapsed solution is the solution to minimize the neural collapse margin. We have modified the expression here in the next version.
>
>   - **The difference between Unconstrained Layer-Peeled Model and Layer Peeled Model**: As discussed in lines 114-116, the difference is that the Layer-Peeled Model includes an explicit constraint on the norm of feature and parameter while the Unconstrained Layer-Peeled Model does not include such a constraint. We hope to justify that the neural collapse phenomenon also emerges even in the absence of explicit constraint.
>
> - **Regarding the claims that the reviewer prefers to have a detailed discussion:**
>
>   - **The claim "the analysis provides insights ...... on generalization and robustness"**: In general, we want to justify how neural collapse emerges in the terminal phase of training. We have proved that, after the data is perfectly separated, gradient flow will gradually find neural collapse solutions that are known to be beneficial to the robustness and generalization of neural networks. A detailed discussion has been made in the original neural collapse paper [2]
>     [2] Prevalence of neural collapse during the terminal phase of deep learning training.
>   - **The claim that our work is stronger than [43]**: As shown in section 3.1, our paper provides an analysis of the training dynamics and shows how gradient flow will converge to those KKT points. In section 3.2, we use second-order analysis to close the gap between neural collapse solutions and KKT points by showing that except for some neural collapse solutions, all KKT points are strict saddle points on the sphere. We include the whole training dynamics in and that's why we believe our results are stronger than [43].
>   - **The claim that our setting stands closer to realistic training**: First we would like to refer to [3] as an empirical justification, they find that neural networks continue to perform well after all the regularizers are removed and provide numerous results to support this. See section 3 in [3] for detailed results and discussion. The feature regularization or constraint is never used in realistic training, but neural networks can still achieve good performance even in the absence of explicit regularization (i.e. weight decay). Except for this practical consideration, it’s hard to plug in a neural network when features are constrained while our setting appears to be more capable for further analysis on real neural networks since we admit the largest freedom of features. By removing explicit feature regularization and constraint, we hope to inspire more theoretical results on the real neural networks.
>     [3] Understanding deep learning requires rethinking generalization.
>
> - **Regarding the assumption that the dataset is balanced:** When the dataset is unbalanced, we would like to refer to [4] where the author characterizes the global optimal solution by minority collapse. In fact, our theoretical analysis only relies on the global optimality of neural collapse and it is not difficult to generalize it to minority collapse. However, the minority collapse does not have a closed-form solution which makes the analysis rather complicated on notation, so we only focus on the balanced case for simplicity and leave the detailed discussion on imbalanced cases for future works.
>   [4] Layer-peeled model: Toward understanding well-trained deep neural networks.
>
> - **Regarding the comments on our experiment:** We also conduct additional experiments on neural networks and results are provided in section D in the appendix, including VGG-13 on CIFAR-10 and ResNet-18 on FashionMNIST and CIFAR-10. The results are consistent with those in section 4 and thus we think our finding is robust to the choice of network architecture and dataset.
>
> - **Regarding the assumption that the neural network trained without weight decay:** As mentioned above, we would like to refer to [3] as an empirical justification that the networks continue to perform well after all the regularizers are removed. In our new version, we have added new experiments to compare the convergence speed in those settings. Generally speaking, training with weight decay is faster than that without weight decay. However, this gap can be closed by incorporating the normalization technique.

---

> > ### Comment · Reviewer_6dN9 · 2021-08-18
> > **Additional question to the authors**
> >
> > Thanks for the clarifications, which furthered my understanding of the theoretical contribution of the paper.
> >
> > I came to a similar conclusion as *Reviewer GFas*, i.e. that the theory does only partially answers  the question"How Gradient Descent Separates Data with Neural
> > Collapse".
> > That is, the theory guarantees convergence once the features are separated ($L<\log(2)$ or $q_{min}>0$). However, if, and due to which effects, the conditions $L<\log(2)$ or $q_{min}>0$ can always be achieved during the optimization is not covered. However, this phase of training is precisely when data gets "separated". \
> > Would you like to comment on this?
> >
> > I observed that the assumption $q_{min}>0$ is missing in Theorem 3.4, whereas it is present in the corresponding Theorem C.2 in the supplementary material. Could you clarify, if the assumption is required for Theorem 3.4.

---

> > > ### Author Response · Authors · 2021-08-19
> > > **Response to additional quesion**
> > >
> > > Thanks for your response. The condition $q_{min}(W, H)>0$ is here to provide an analysis of how gradient flow converges to the first order KKT point. Actually, the landscape analysis can be generalized to the early time setting easily. To see this, please check out the proof of Theorem C.2. in the appendix. The condition $q_{min}>0$ is used to guarantee $\lambda<0$ in lines 229-230. However, when $\lambda>0$, by equation (73) we know $\frac{d^2}{dt^2}h(\phi(t))=0$, you can change the minus sign in equation (75) to be a plus sign. Then just change the $-\lambda$ to be $\lambda$ in equations between lines 242-243, lines 246-247 and in line 243, all of our analysis holds still. The remaining case is $\lambda = 0$. When $\lambda = 0$, we even do not need the analysis between lines 240-243, equations between lines 246-247 only are enough to find a decreasing direction since $\nabla L(WH)$ is of rank $n-1$.
> > >
> > > As a result, our landscape analysis in section 3.2 can be generalized to the whole parameter space and show that neural collapse solutions are unique minimizers. One can conclude that gradient descent with stochastic perturbation will find neural collapse solutions since the stochastic gradient descent can escape saddle points. But with the help of assumption $q_{min}(W, H)>0$​, we can have a more refined analysis about how gradient flow converges in direction and obtain an exact convergence rate (the proof is similar to [1] it will be included in our next version). How to generalize this part into early time dynamics is an interesting problem and we will continue to work on this, but currently, this assumption is necessary for all of the implicit bias analysis in the nonlinear setting [1] [2] [3]. We must apologize for bringing this unnecessary confusion (we merged the assumptions of the two-section in the first version and leads to the unnecessary assumption for the landscape analysis).
> > >
> > > Regarding the difference between Theorem 3.4 and Theorem C.2, we must apologize for this mistake. Actually, this condition is not necessary for Theorem 3.4, and the reason has been stated above.
> > >
> > > It’s glad to find our clarification contributes to a further understanding for the reviewer. We hope you will also be satisfied with our response above and convinced to increase the score, helping our paper to be accepted. Please do let us know if you have any other concerns.
> > >
> > > [1]Kaifeng Lyu and Jian Li. Gradient descent maximizes the margin of homogeneous neural networks. arXiv preprint arXiv:1906.05890, 2019.
> > >
> > > [2]Ji, Ziwei, and Matus Telgarsky. "Directional convergence and alignment in deep learning." *arXiv preprint arXiv:2006.06657* (2020).
> > >
> > > [3]Nacson, Mor Shpigel, et al. "Lexicographic and depth-sensitive margins in homogeneous and non-homogeneous deep models." *International Conference on Machine Learning*. PMLR, 2019.

---

> > > ### Author Response · Authors · 2021-08-21
> > > **Looking forward to the response.**
> > >
> > > We wonder if our answer addresses your concern. If the reviewer still has questions, we’d love to have a further explanation. If the concerns are addressed, we hope the reviewer can improve the evaluation of our paper.

---

> > > > ### Comment · Reviewer_6dN9 · 2021-08-21
> > > > **The response**
> > > >
> > > > To be honest, I could not really verify, if the condition q_min>0 could be removed from the theorem. This is because, in its current form the proof (similar to the paper) is difficult to follow due to the grammatical errors. Sometimes I needed to guess, what was actually meant in the statements. However, from my perception, it might indeed be possible to remove the assumption.
> > > >
> > > >
> > > > Unfortunately, these calrity and linguistic issues are a common theme in the paper, and, to large extend, impaire your contribution.
> > > > As you stated, you are already working on improving the presentation. Unfortunately, due to the specifics of the review process, we cannot inspect this revision. Considering the scope of these issues, I doubt that they will be cleared to a satisfactory level and I cannot with good conscience recommend acceptance of the paper. I want to empahsize, that in general, I am in favour of the paper and that I would like to see it published at some point, but I feel that the revison would need another round of reviews.

---

> > > > > ### Author Response · Authors · 2021-09-06
> > > > > **Still waiting for response**
> > > > >
> > > > > Dear reviewer:
> > > > >
> > > > > The decision deadline is coming, and we are still eager to hear from the reviewer. We are willing to discuss if the reviewers have further concerns but it seems that the communication is blocked. Whether you update your score may be critical to a fair decision on our paper.
> > > > >
> > > > > Actually, it's very easy to show how this condition can be removed. In our proof, you can find that the only need for this condition appears in line 230 and we show it's equivalent to $\lambda<0$. We then use this condition to derive the equation (77) $\|W\|_2=\|H\|_2$ and $\|W\|_F=\|H\|_F$ from equation (76), you can divide \lambda on both hands of the equation and of course it holds when $\lambda>0$. Also in equations between lines 242 and 243, we only take the norm on $\lambda W$ and of course, you can change this $-\lambda$ to be $|\lambda|$ to cover the $\lambda>0$ cases. Then everything holds immediately in the equation between lines 246-247 and we find a decreasing direction again. (Note that the $-\lambda$ can be changed into $\lambda$ because the second term equals to zero always by equation 73) So the remaining case is just $\lambda=0$, which is much easier since it implies that $\nabla L(WH)H^T=0$. Now just focus on the equation between lines 246-247, the second term disappears because $\lambda=0$ and a decreasing direction always exists by the construction because $ \nabla L(WH)$, as we have shown in line 214, is of rank $K-1$​.
> > > > >
> > > > > We must apologize for this confusion caused by the linguistic errors again. We understand that the reviewer may feel uncomfortable about them thus is not willing to give a clear acceptance to our paper (7: Good paper, accept), but we believe those linguistic errors do not hide the proof ideas, impair contributions of our work, and lead to a "4: OK but not good enough" score. At the same time, Reviewer XiQd thinks our paper is “easy to follow”，Reviewer HKcz agrees that “our proof technique is clear”. Under this situation, we hope that this paper can be reevaluated and not be rejected just because of linguistic errors. Actually, we consider giving a "4: OK but not good enough" score just because of linguistic errors as an **unfair** decision.
> > > > >
> > > > > The decision deadline is coming, we are here still waiting for the response from the reviewer. It's sad to see the communication is blocked during the NeurIPS review process.

---

### Decision · Program_Chairs · 2021-09-27

**Decision:**

Reject

**Comment:**

This paper studies an asymptotic implicit bias phenomenon termed "neural collapse".  It constitutes a very interesting line of work, however reviewers overall had many concerns, and especially during internal discussion it was highlighted that the paper is very hard to follow and feels unpolished in many places.  Therefore I urge the authors to continue their work, but to be attentive to reviewer concerns, especially on presentation clarity.